# Modeling the Influence of Okara Flour Supplementation from Time-Temperature Drying Treatment on the Quality of Gluten-Free Roll Produced from Rice Flour

**DOI:** 10.3390/foods12183421

**Published:** 2023-09-14

**Authors:** Pavalee Chompoorat Tridtitanakiat, Zorba J. Hernández-Estrada, Patricia Rayas-Duarte

**Affiliations:** 1Division of Product Development Technology, Faculty of Agro-Industry, Chiang Mai University, Chiang Mai 50100, Thailand; pavalee.t@cmu.ac.th; 2Robert M Kerr Food & Agricultural Products Center, Department of Biochemistry and Molecular Biology, Oklahoma State University, Stillwater, OK 74078, USA; zorba.he@veracruz.tecnm.mx; 3Lanna Rice Research Center, Chiang Mai University, Chiang Mai 50200, Thailand; 4Tecnológico Nacional de México/I.T. Veracruz, Calz. Miguel Angel de Quevedo 2779 Col. Formando Hogar, Veracruz 91860, Mexico

**Keywords:** Okara, burgers model, gluten-free product, drying process

## Abstract

Okara, an unassuming residue, is emerging as a notable reservoir of essential nutrients, encompassing an abundant supply of protein, dietary fiber, and potent antioxidant components. Hence, the incorporation of okara as an ingredient in the production of rice flour-based rolls held a considerable interest in nutritional and functional aspects. Okara flour supplement was prepared by drying at 100 °C for 2 h and selected based on the highest antioxidant level. Gluten-free rolls were prepared containing 0, 5, and 10% okara flour dried at 100 °C for 2 h, and their physicochemical properties were analyzed. Okara flour addition reduced the deformation of gluten-free batter roll by improving solid and liquid-like behavior, as evaluated with rheological measurements. Rolls containing okara flour processed at 100 °C for 2 h had increased firmness and decreased specific volume compared to the control. However, there were no significant differences in the sensory evaluation scores, suggesting that the consumers’ acceptance of the control and the Okara rolls was similar. Okara flour supplement at 10% addition led to the nutritional improvement of the gluten-free rolls (increase of 2.4% protein and 1.32 times dietary fiber, dry basis). In contrast, there were no significant differences in the antioxidant level compared to the control. Okara flour is a functional ingredient with potential use in gluten-free products and a variety of novel products where enrichment is desirable.

## 1. Introduction

Okara, soy pulp (*Glycine max* (L.) Merr.), is a byproduct from soybean products such as soy milk and tofu that contains essential nutrients associated with health [1,2,3,4]. Each 100 kg of soymilk production yields an equal or larger amount of okara that is normally used for animal feed as industrial waste [5]. The high content of protein (15.2–33.4%) and fiber (42.4–58.1%) makes okara a potential functional ingredient [1]. Furthermore, okara contains isoflavones, a group of heterocyclic phenols with health benefits, including the prevention of cardiovascular diseases, cancer, and menopausal symptoms [1]. Soybean is known as a source of antioxidants such as flavonoids and isoflavones. Food processing methods could impact isoflavones content. There were six main components of isoflavones with medicinal properties in okara, which are daidzin, glycitin, genistin, daidzein, glycitein, and genistein. The genistin was the primary okara glucoside according to a high-performance liquid chromatography study. Examining thermal treatment effects on drying okara flour, it was found that higher temperatures (200 °C) yielded increased levels of isoflavones (genistin, genistein, daidzin, and daidzein) in comparison to drying at 80 °C [6]. The study on the antioxidant activity of okara showed that the okara (soluble polysaccharide fractions) had in vitro reduction power and free radical scavenging of 11–26 μmol Trolox Equivalent/g dry weight and 63–78 μmol TE/g dry weight, respectively [7]. Thus, it is interesting to study the impact of drying and breadmaking processes on the antioxidant level in okara flour and its applications. 

Dietary fiber can aid in the control of chronic disease such as diabetes, cancer, and coronary heart diseases [8]. Dietary fiber is the edible parts of plants or carbohydrates that are resistant to digestion and absorption in the small intestine, with partial fermentation in the large intestine [9]. It can also help provide a potential benefit in increasing bowel movement. Consequently, increasing the level of dietary fiber intake is currently a healthy trend, especially in bakery products [9]. In breadmaking, insoluble dietary fiber changes the rheological properties of dough (i.e., dough viscosity or consistency), leading to high bread quality [10]. With its ability to increase water and fat binding, this can also help improve loaf volume and increase bread softness during storage. In addition, the partial replacement of flour by fiber in bread can also provide a lower carbohydrate level with reduced caloric content. Soy okara flour was also revealed to enhance the viscosity of gluten-free batter and moistness of gluten-free bread [11]. Moreover, incorporating okara into pasta fortification revealed no structural alteration, indicating the high potential of okara flour as a functional ingredient in the food industry [12]. However, limited research has observed variations in okara’s antioxidant levels during processing. Currently, the demand for gluten-free products catering to individuals with celiac disease or wheat allergies is on the rise. In gluten-free products, wheat flour is normally substituted with high-carbohydrate white rice flour. Therefore, in this study, we focused on the processing of soybean pulp (okara) to produce a novel functional ingredient in gluten-free rolls.

The objective of this study was to investigate the effect of temperature-time drying treatments on okara flour’s chemical properties and the incorporation of okara flour in gluten-free rolls. 

## 2. Material and Methods

### 2.1. Okara Flour Preparation

Okara was obtained from soymilk production. The soymilk process began with soaking soybeans at 5:1 ratio water/soybean for 4 h. The 300 g of hydrated soybean was then blended with extra water (300 mL) using a blender (KitchenAid, St. Joseph, MI, USA) for 3 min. The soybean pulp (okara) was filtered with a double-layer cheesecloth. The okara was dried with three temperature-time treatments: 70 °C for 4 h, 80 °C for 3 h, and 100 °C for 2 h; the treatments were selected from a preliminary study. Dried okara was ground in an electric grain mill (Blendtec, Orem, UT, USA), screened to pass a 250 μm sieve particle sizes (60 US mesh screen), and stored in sealed polyethylene bags until needed for analyses. 

### 2.2. Chemical Properties of Okara Flour

Each temperature-time treatment of okara flour was analyzed following AACC Approved Methods of Analysis for moisture (44–17.01) [13], ash (08–16.01) [14], protein (46–13.01) [15], fat (30–25.01) [16], total dietary fiber (32–05.01) [17], and carbohydrate by difference. Antioxidant level was analyzed according to the DPPH method expressed as μmol TE (Trolox equivalents)/g based on dry weight [12]. Briefly, an extraction of 20 g of okara flour with 200 mL methanol:water (80:20, *v*/*v*) mixture was done. The supernatant (0.1 mL) from centrifugation (10,000× *g* for 15 min at 5 °C) was used to react with 0.1 mM methanolic DPPH solution (3.9 mL). Then, the assay mixture was incubated and shaken for 30 min at ambient temperature in a dark place. The mixture sample was analyzed for an absorption at 517 nm using UV/Vis Spectrophotometer (Lambda 25, Perkin Elmer, Waltham, MA, USA) [12]. In addition, the total microbial count of okara flour was also analyzed for safety purposes.

### 2.3. Preparation of Gluten-Free Roll 

A commercial rice flour (Cho Heng Rice Vermicelli Factory Co., LTD. Bangkok, Thailand) was used for gluten-free rolls. The rice flour composition was 6.9% protein, 1.3% fat, 0.5% total dietary fiber, and 79.8% carbohydrate by difference. Okara flour from the 100 °C and 2 h treatment was selected for its highest level of antioxidants, whereas the protein and total dietary fiber were not significantly different in the three treatments (Table 1). Okara flour was added at 0 (control), 5, and 10% based on rice flour. The gluten-free roll ingredients were rice flour 100%, gum 2.4%, yeast 1.9%, sugar 15.6%, salt 1.6%, water 60.3%, oil 11.8%, and egg 32.4%, based on rice flour, and remained the same in all treatments with the addition of okara flour. The preparation of gluten-free batter followed a typical straight-dough procedure in bread. Dry ingredients were mixed at low speed (60 rpm) for 3.5 min (KitchenAid, St. Joseph, MI, USA). Then, wet ingredients were added and mixed for 3 min at medium speed (111 rpm). A sample of 130 g was rounded in a pan using a spatula, proofed for 60 min at room temperature, and baked at 200 °C for 22 min in a convection oven (Electrolux EOG 601 X Gas single oven). During proofing, the batter was covered with plastic wrap to prevent dryness. The mixer with a flat-beater attachment was used in this study. Breadmaking tests were performed in three independent experiments. 

### 2.4. Chemical Properties of Gluten-Free Roll

Proximate analyses of gluten-free roll with okara were performed with AACC Approved Methods for moisture content (44-31.01) [13], ash (08-01.01) [14], fat (46-30-25.01) [15], protein (46-30.01) [16], and total dietary fiber (46-32-05.01) [17]. The oxygen radical absorbance capacity in the gluten-free roll with okara was determined and expressed as μM of Trolox equivalents per 100 g of sample [12]. 

### 2.5. Small Deformation of Gluten-Free Roll Batter 

#### 2.5.1. Frequency Sweep Test 

Rheological measurement of gluten-free roll batter without yeast addition was conducted by frequency sweep test using a rheometer AR-1000N (TA Instruments, New Castle, DE, USA) equipped with a Peltier plate controlling the temperature to 25 °C. The parallel plate (25 mm cross-hatched geometry) was lowered down to a batter sample with a 1 mm gap [18]. The batter edge was trimmed using the spatula before covering with the trap to prevent changes in structure due to edges drying out. The batter sample was rested for 1 min before testing with a dynamic oscillatory test. The test was set for 0.1–10 Hz at a constant strain (0.5%) within the linear viscoelastic region. Parameters obtained from this test were G’ (elastic or storage modulus), G” (viscous or loss modulus), and tan γ (tan delta or ratio of G” to G’. This test was performed in three independent samples [18].

#### 2.5.2. Creep-Recovery Test

The sample was prepared as described in Section 2.5.1 for the frequency sweep test. After the batter sample was transferred to the lower plate of the rheometer AR-1000N (TA Instruments, New Castle, DE, USA), it was enclosed in the moisture trap for 1 min under a parallel plate (25 mm cross-hatched geometry) with 1 mm gap to allow the structure to recover from the mixing process. Then, the test was conducted in creep-recovery mode. The creep phase consisted of applying a constant shear stress (20 Pa) into a batter sample within each sample’s linear viscoelastic region (LVE) for 100 s. Next, the rheometer released the shear stress (0 Pa) to allow the batter to recover after shearing for another 100 s. Thus, the total time measurement was 200 s [18]. Compliance *J* (Pa^−1^) was measured for deformation during the creep and recovery test as a function of time. Compliance is a ratio of recorded creep strain (γ) to constant shear stress (σ) or equal to 20 Pa for this study, as shown in Equation (1).
(1)$$J_{\left(t\right)}=\frac{\gamma_{\left(t\right)}}{\sigma }$$

The experimental data obtained from this test was fitted into a 6-parameter Kelvin-Voigt model (Equation (2) for creep phase and Equation (3) for recovery phase) [19]. During creep or retardation phase, ${Jc}_{o}$ parameter reflects the spring element, which could be related to the elasticity of the material. ${Jc}_{1}$ and ${tc}_{1}$ reflect the combination of spring and dashpot, which could be related to the viscoelastic properties of the material for the first element. ${Jc}_{2}$ and ${tc}_{2}$ reflect the combination of spring and dashpot, which could be related to the viscoelastic properties of the material for the second element. $\eta_{o}$ reflects pure dashpot, which could be related to pure viscosity. For the recovery phase, the explanation of each element is similar to those of the creep phase. However, there is no pure dashpot or viscous flow during the recovery phase.
(2)$$J_{c}\left(t\right)={Jc}_{o}+{Jc}_{1}\left(1-exp(\frac{-t}{{tc}_{1}})\right)+{Jc}_{2}\left(1-exp(\frac{-t}{{tc}_{2}})\right)+\frac{t}{\eta_{o}}$$
(3)$$J_{r}\left(t\right)={Jr}_{o}+{Jr}_{1}\left(1-exp(\frac{-t}{{tr}_{1}})\right)+{Jr}_{2}\left(1-exp(\frac{-t}{{tr}_{2}})\right)$$

### 2.6. Physical and Textural Properties of Gluten-Free Rolls

#### 2.6.1. Volume and Color of Gluten-Free Rolls and Batter

Gluten-free rolls with three different levels of okara were prepared as described in Section 2.3. Each gluten-free roll batter was weighed before baking and after baking for 1 h (loaf weight, g) to record yield loss (%). Loaf volume (cm^3^) was measured using rapeseed displacement. Specific volume (cm^3^/g) was also calculated using loaf volume (cm^3^) divided by loaf weight (g). Crumb color of gluten-free roll and batter were also analyzed with a HunterLab ColorFlex (Hunter Associates Inc., Reston, VA, USA). Parameters of L* (brightness; 0: black, 100: white), a* (+a: redness; −a: greenness), and b* (+b: yellowness; −b: blueness) values were recorded [19]. Measurements were recorded in three replicates.

#### 2.6.2. Texture of Gluten-Free Rolls

Gluten-free rolls were cooled for 1 h after baking, sliced into 1 cm thickness, and stored in ziplock plastic bags. The gluten-free roll crumb firmness was measured by texture profile analysis (TPA) using a TA-XT2 texture analyser (Stable Microsystems, Surrey, UK) (50 kg load cell) with a 25 mm diameter cylindrical probe. The conditions were set as pre-test speed, 2.0 mm/s; crosshead speed, 1 mm/s; post-test speed, 5.0 mm/s and compression to 40% height [20]. Measurements were conducted in two replicates per treatment and 12 subsamples per replicate.

### 2.7. Sensory Properties of Gluten-Free Rolls

Sensory analysis of gluten-free rolls was performed with a consumer acceptance test the day after baking (day 1 of storage) [18]. The samples were evaluated by 25 untrained panelists, students from Maejo University, Thailand, aged 19 to 25, of which 40% were male and 60% female. The gluten-free roll samples were assigned random three-digit code numbers and presented as 1.5 cm thick slices, with a total of five samples evaluated per session. Sensory attributes of texture, flavor, aroma, and overall acceptability were scored using a nine-point hedonic scale with 1, 5, and 9 representing extremely dislike, neither like nor dislike, and extremely like, respectively. 

### 2.8. Statistical Analysis

The experimental design for this study was a completely randomized design (CRD). Analysis of variance (ANOVA) procedures were used, assuming a model in a completely randomized design using the SAS program (Version 9.1 SAS Institute Inc., Cary, NC, USA). The mean significant difference was tested using Tukey’s multiple comparison test (α = 0.05). The multiple analysis by ordination called principal component analysis (PCA) using Canoco for Windows 5 software (Centre for Biometry, Wageningen, The Netherlands) was applied to measure correlation among parameters and samples [21]. 

## 3. Results and Discussion

### 3.1. Okara Flour Characteristics

The properties of okara flour at different temperature levels and times were shown in Table 1. The results revealed that temperature-time drying treatments of okara impacted some of the chemical properties of okara, which were moisture content, ash, total dietary fiber, total antioxidant, and total plate count (*p* < 0.05); in contrast, there was no significant difference in carbohydrate, fat, and protein (dry basis). The total dietary fiber and moisture content were lower after heating at the higher temperature for a shorter time (*p* < 0.05). A previous study found that using the 40% power level of a microwave oven to dry okara flour could also reduce the moisture content of okara up to 1 h for processing. The beginning of the water evaporation process happened when the okara temperature reached 91 °C [22]. In this present study, it took up to 2 h using an oven temperature of 100 °C to yield okara flour with 7% moisture content and the highest level of total antioxidant. Different dried methods, such as forced-air oven drying, microwave drying, and freeze drying, can cause a change in dietary fiber [23]. The average of total dietary fiber, insoluble fiber, and soluble fiber of okara from literature reports were 54.6, 49.4, and 4.9 g/100 g dry basis, respectively. Thus, the insoluble-to-soluble fiber ratio in okara can be assumed as 10:1 (*w*/*w*). Using 100 °C for 2 h for drying okara flour yielded the highest level of total antioxidant (1.35 times higher than using 70 °C for 4 h) (*p* < 0.05). Thus, the combination of time exposure to heat and temperature level significantly affected the antioxidant level of okara flour. The study of the total phenolic content (TPC) of microwave-processed pasta with 10–50% of okara flour showed that the TPC can be varied, from 158.37 to 232.90 mg (GAE/100 g) [24]. In a similar trend, the total plate count was significantly lower after drying at 80 °C for 3 h and 100 °C for 2 h compared to 70 °C for 4 h, up to 48.3% (*p* < 0.05). 

### 3.2. Chemical Properties of Gluten-Free Roll 

Okara flour obtained from temperature-time treatment of 100 °C for 2 h was selected for use in gluten-free rolls based on the highest antioxidant levels and shortest drying time. Okara flour was incorporated in different levels (0, 5, 10% based on rice flour) into gluten-free rolls to improve the nutritional value. Using 10% okara flour in gluten-free rolls significantly increased total dietary fiber (1.32 times, dry basis) and protein content (2.4%, dry basis), compared to control (0% okara flour addition) (Table 2). The total dietary fiber was improved from 3.23 to 7.5% (dry basis) and the protein from 8.3 to 8.5% (dry basis) (*p* < 0.05). Thus, the gluten-free rolls could be claimed as high-fiber products (according to conditions for nutrient content claims for dietary fiber from FAO). However, adding okara at 5% in the gluten-free roll did not improve protein content. There were no significant differences in total antioxidant in gluten-free rolls containing up to 10% okara flour compared to the control (8.1 mg Trolox/100 g, dry basis) (Table 2). This suggests that a significant addition of okara flour would be needed in the rolls to make a significant contribution. However, using supplementation of 10% okara in bread was found to increase in caloric content due to a higher percentage of fat and protein from okara [4]. Thus, the selection of okara flour level must be carefully chosen depending on a purpose of use. 

### 3.3. Gluten-Free Roll Batter Rheological Properties

The Burgers model has been applied to study rheological changes related to the internal structure and changes in gluten-free cake batter systems using mechanical models [24,25,26,27,28,29]. Typical creep-recovery curves for values of compliance as a function of time are depicted in Figure 1. The effect of okara levels on the creep behavior of the batter of the gluten-free rolls was recorded in the time interval between 0 and 100 s; recovery behavior was recorded in the interval between 100 and 200 s. The viscoelastic properties based on the creep-recovery test of the gluten-free roll batter with and without okara had typical viscoelastic behavior similar to gluten-free batter, gluten, and dough properties [24,30,31,32,33,34]. The modeled parameters from the Burgers model of gluten-free roll batter with and without okara are shown in Table 3. The fitting of compliance as a function of time based on the Burgers model for this gluten-free roll batter system with and without okara had values of r-square higher than 0.99 in all samples (data are not shown). $J_{max}$ (Final value of compliance during creep) and $J_{final}$ (Final value of compliance during recovery) are experimental parameters from this measurement as well as recoverability (RCY), which is calculated by Equation (4). The flowability of material can also be determined by subtracting final compliance from maximum compliance ($J_{max\;}-J_{final})$.
(4)$$\mathrm{RCY}\;(\%)=(J_{final}*100\;/\;J_{max})$$

The addition of okara to gluten-free roll batter resulted in a significant decrease in deformation, increase in recoverability, and decrease in flowability ($J_{max}$ – $J_{final}$) (*p* < 0.05) (Table 3). The Burgers model parameters revealed that instantaneous and retarded deformation behavior of gluten-free roll batter significantly decreased by 51–60% in all elements during creep and recovery phenomena compared to gluten-free batter roll without okara (*p* < 0.05). Thus, the addition of okara strengthens the structure of gluten-free batter rolls. Zero shear viscosity improved significantly after adding okara by up to 70.6% (*p* < 0.05). There was a significant decrease in retardation time in both creep and recovery phases by up to 18.7% ($J_{r2})$ (*p* < 0.05). The elements of the Burgers model could be attributed to the entanglement of long-chain polymers in the material structure. The addition of okara flour and its consequent increase in total dietary fiber and protein could reinforce the batter structure of gluten-free rolls by increasing the hydrogen bonding with water around its particles. Okara insoluble fiber structures normally consist of cellulose, hemicellulose, and lignin [34]. The study of interactions between water molecules and cellulose polymers showed that water tended to bond strongly with cellulose through capillary attraction, which led to an increase in water absorption [35]. 

The viscoelastic properties of all gluten-free batter rolls were frequency dependent (data not shown). Storage modulus (G’) was higher than loss modulus (G”) in all samples throughout the frequency range, suggesting that gluten-free batter rolls with and without okara had solid-like properties (Figure 2). An increasing level of okara in gluten-free batter rolls also led to an increase in solid-like behavior, with a higher increase in storage modulus (G’) (113.8%) compared to loss modulus (G”) (97.5%) with 10% okara flour (*p* < 0.05) (Table 3). In addition, the magnitude of both moduli increased as the range of frequency increased. This means that the strength of gluten-free batter increased at a higher frequency during the frequency sweep test. The high dietary fiber content in okara suggests a high water-absorption capacity and thus, influences the batter’s rheological properties and the gluten-free roll’s properties. This test was performed by varying frequency while deformation or shear stress of amplitude remained unchanged. It is possible that the batter formation improved with better water absorption during higher frequency, and the batter structure became more organized by hydrogen bonds. 

### 3.4. Gluten-Free Roll Properties

Texture profile analysis (TPA) is a physical measurement designed for mimicking mouth biting action by compressing a sample twice in order to evaluate the sample as it is chewed. The results showed that okara flour increased hardness of gluten-free crumb rolls by up to 155% at 10% okara flour addition; on the other hand, the springiness was lower by up to 53.6% compared to gluten-free crumb rolls without okara (*p* < 0.05) (Table 3). There was no significant difference in cohesiveness and resilience. The results were in agreement with a study of okara flour in wheat bread [10,22]. The impact of okara in this previous study could have been caused by gluten dilution, the formation of soybean’s fiber in the gluten structure, and the disulfide bond exchange between gluten and soy protein. All of these factors affected dough viscosity, leading to lesser firmness, cohesion, and chewability of the bread with okara. Thus, the increase in viscosity of this gluten-free roll batter could affect the hardness of gluten-free crumb as well. It was also found to have an adverse effect on the quality of products enriched with okara powder. A study of stress relaxation on cooked noodles with okara revealed that noodles with okara powder at 10–15% had lower elasticity than those with 0–5% [35].

Gluten-free roll samples enriched with okara flour were tested, and their characteristics are reported in Table 3. Okara decreased volume, specific volume, and yield loss at 5% and 10% okara addition, compared to control (*p* < 0.05). In agreement with the study of okara incorporation in gluten-free cookies, a decrease in the specific volume of gluten-free cookies was obtained after adding okara in higher amounts [36]. The impact of protein and fiber might interfere with the structure matrix and reduce gas retention capacity in the batter system. Interestingly, a study of substitution of wheat flour with okara flours in sweet biscuits showed no significant difference in volume of the biscuit at 30% substitution [37]. Generally, the final product of the bread-like product with high amounts of insoluble fiber had an improvement in volume and height because the insoluble fiber increased dough viscosity and helped entrap the gas during fermentation [8]. Thus, okara flour had a different impact on the gluten-free batter roll and biscuit wheat dough systems. The color of batter and rolls was significantly darker after the addition of okara at 5% and 10% compared to 0% addition (*p* < 0.05). In addition, the overall characteristics of gluten-free rolls with okara flour (dried at 100 °C for 2 h) were evaluated using the hedonic test and are shown in Table 3. The overall characteristics of the rolls were similar, with a score range of 6.9–7.0. All formulations were acceptable since the untrained consumers gave a score higher than 5, which represented neither like nor dislike. Thus, the results indicated that okara flour could be used at the higher addition level (10%) in the gluten-free rolls to improve its nutritional value. 

### 3.5. Principal Component Analysis (PCA)

The biplot graph shows the separation between samples from gluten-free rolls with different levels of okara flour in a very high variation with 99.6% total explained variance (Figure 3). The principal component 1 (PC1) was the main axis in this study, which contained all the physical properties of gluten-free roll and batter containing okara flour at 0, 5, and 10%, with 97.3% explained variance. The main contributors were springiness, pure viscosity, lightness, and yield loss; these vectors are aligned closely to axis 1. The compliance parameters known as deformation parameters were opposite to the solid and viscous-like parameters and pure viscosity variables. The gluten-free roll batter with 0% okara flour was closely related to the springiness of the roll and deformation parameters. When adding a higher amount of okara flour, the samples moved towards hardness of rolls and pure viscosity, as well as solid and viscous-like parameters of batter. Incorporating okara flour into gluten-free cake similarly heightened batter density, viscosity, and cake hardness [38]. Relaxation time (*t_c_*_1_ and *t_r_*_2_) were independent of other parameters; on the other hand, tr1 moved closely to deformation parameters. The result of relaxation time showed the same trend as the modeling in gluten-free red kidney bean cupcake batter [19]. The overall acceptability, shown as sensory, revealed a high correlation to the solid and viscous-like parameters of batter. The organoleptic parameter (sensory) was in the same quadrant as the gluten-free roll with 5% okara addition. It is noticeable that the distance between the sensory vector and samples of 5% and 10% okara was similar. Thus, the results from Tukey’s multiple comparison test in Table 3 showed no significant difference in the overall quality of gluten-free rolls with 5% and 10% okara. In addition, 10% okara addition was closely related to the hardness and gumminess of gluten-free rolls. 

## 4. Conclusions

Okara dried at 100 °C for 2 h showed the highest antioxidant value, which was 26.2% more than drying at 70 °C for 4 h. Thus, this process condition was chosen for drying fresh okara and making gluten-free rolls. Using okara flour at 0, 5, and 10% revealed that okara flour impacts the viscoelastic properties of batter as well as the physical properties of a gluten-free roll. The gluten-free roll batter had lower flowability and deformation after adding okara flour at 5 and 10% compared to control. The gluten-free roll with 10% okara had the highest hardness crumb and the lowest value in specific volume. The okara treatment of 100 °C for 2 h produced the highest antioxidant level but did not survive the baking process of the gluten-free rolls with the addition of 10% of okara flour. A supplementation of okara at a higher percentage with the incorporation of biopolymers should be focused on in future work toward food-waste reduction efforts. 

## Figures and Tables

**Figure 1 foods-12-03421-f001:**
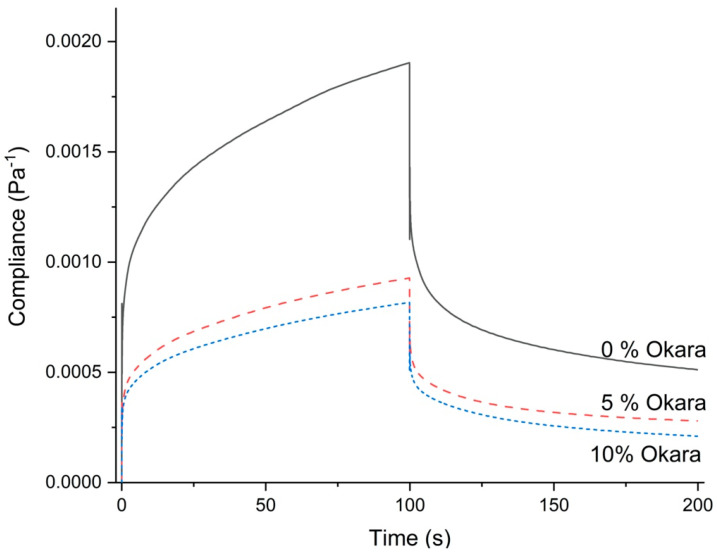
Compliance of gluten-free roll batter with okara flour addition at different levels as a function of time.

**Figure 2 foods-12-03421-f002:**
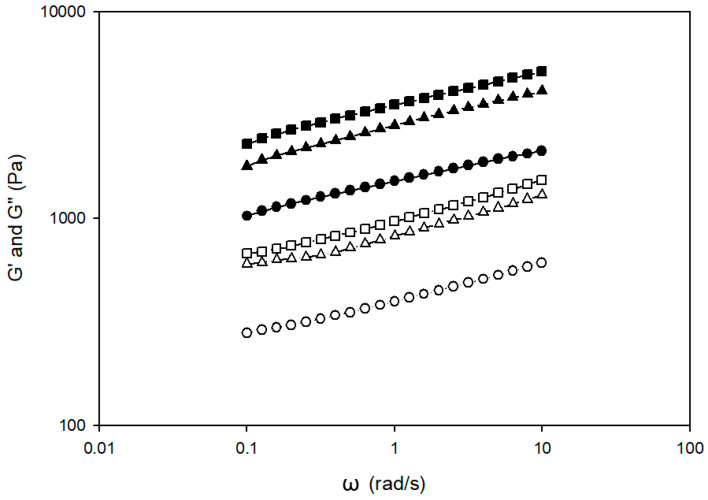
Storage (G’) and loss (G”) modulus as a function of frequency (ω) of gluten-free batter roll with okara flour addition at different levels. Symbols represent levels of okara flour addition: 0%, filled (G’) and open (G”) circle; 5% filled (G’) and open (G”) triangle; and 10% filled (G’) and open (G”) square.

**Figure 3 foods-12-03421-f003:**
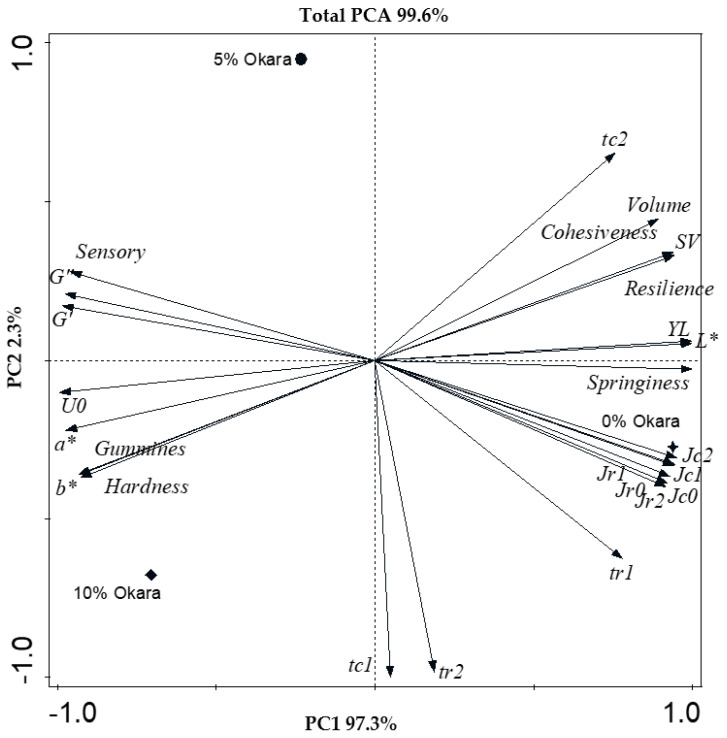
Biplot graph of principal component analysis (PCA) of parameters from fundamental rheological properties, physical properties, and overall sensory score showing the effect of okara addition (definitions in Table 3). Symbols: 0% okara, filled star; 5% okara, filled circle; and 10% okara, filled diamond.

**Table 1 foods-12-03421-t001:** Effect of time-temperature drying treatments on okara flour proximate analysis with total antioxidant and total plate count *.

Analyses	Time-Temperature Drying Treatment					
	4 h-70 °C		3 h-80 °C		2 h-100 °C	
	As Is	d.b.	As Is	d.b.	As Is	d.b.
Moisture (g/100 g)	6.29 ^a^	-	5.68 ^b^	-	5.28 ^b^	-
Protein (g/100 g)	41.9 ^a^	44.7 ^a^	41.9 ^a^	44.5 ^a^	38.0 ^a^	40.1 ^a^
Lipid (g/100 g)	16.2 ^a^	17.3 ^a^	15.3 ^a^	16.2 ^a^	15.4 ^a^	16.2 ^a^
Ash (g/100 g)	2.80 ^a^	3.00 ^a^	3.09 ^a^	3.40 ^a^	3.07 ^a^	3.31 ^a^
Total dietary fiber (g/100 g)	26.1 ^a^	27.7 ^a^	25.4 ^a^	26.4 ^a^	25.0 ^a^	25.3 ^a^
Carbohydrate (g/100 g)	32.8 ^a^	35.0 ^a^	33.6 ^a^	35.6 ^a^	36.3 ^a^	38.3 ^a^
TPC (mg Trolox/100 g)	26.2 ^c^	27.9 ^c^	28.2 ^b^	29.9 ^b^	35.9 ^a^	37.8 ^a^
Total bacteria count (CFU/g of sample)	1.2 × 10 ^5a^	-	5.4 × 10 ^5a^	-	6.2 × 10 ^4b^	-

* Means (n = 3) with different subscript letters within a row are significantly different (*p* < 0.05, Tukey test). As is = as is basis; d.b. = dry basis. CFU = colony forming units.

**Table 2 foods-12-03421-t002:** Effect of okara flour addition on gluten-free rolls chemical properties *.

Analyses	Okara Flour Addition					
	0% Flour		5% Flour		10% Flour	
	As Is	d.b.	As Is	d.b.	As Is	d.b.
Moisture (g/100 g)	38.4 ^a^	-	39.0 ^a^	-	38.4 ^a^	-
Protein (g/100 g)	5.1 ^b^	8.3 ^b^	5.1 ^b^	8.3 ^b^	5.3 ^a^	8.5 ^a^
Lipid (g/100 g)	8.1 ^a^	13.0 ^a^	7.3 ^a^	12.0 ^b^	7.6 ^a^	12.4 ^b^
Ash (g/100 g)	1.09 ^a^	1.77 ^a^	1.05 ^a^	1.72 ^a^	1.06 ^a^	1.72 ^a^
Total dietary fiber (g/100 g)	1.99 ^b^	3.23 ^b^	2.36 ^b^	3.8 ^b^	4.63 ^a^	7.5 ^a^
Carbohydrate (g/100 g)	47.4 ^a^	76.9 ^a^	47.5 ^a^	77.9 ^a^	47.6 ^a^	77.3 ^a^
TPC (mg Trolox/100 g)	5.01 ^a^	8.1 ^a^	5.02 ^a^	8.2 ^a^	5.00 ^a^	8.1 ^a^

* Means (n = 3) with different superscript letter within a row are significantly different (*p* < 0.05, Tukey test). As is basis; d.b. = dry basis.

**Table 3 foods-12-03421-t003:** Parameters from creep-recovery test, frequency sweep test, texture profile analysis, baking performance test, and color and sensory analysis of gluten-free batter and baked rolls with okara flour addition *.

Measurements	Parameters	Okara Flour Addition		
		0%	5%	10%
**Creep-recovery test of batter**	$J_{max}$ (Pa^−1^ × 10^−3^)	2.0 ^a^	1.3 ^b^	1.0 ^b^
	$J_{final}$ (Pa^−1^ × 10^−3^)	1.6 ^a^	0.6 ^b^	0.7 ^b^
	RCY (%)	67.1 ^c^	71.1 ^b^	76.2 ^a^
	$J_{c0}$ (Pa^−1^ × 10^−5^)	6.2 ^a^	2.1 ^b^	2.6 ^b^
	$J_{c1}$ (Pa^−1^ × 10^−5^)	4.5 ^a^	1.8 ^b^	1.9 ^b^
	$t_{c1}$ (s)	0.9 ^a^	0.9 ^a^	1.0 ^a^
	$J_{c2}$ (Pa^−1^ × 10^−5^)	5.0 ^a^	2.1 ^b^	2.2 ^b^
	$t_{c2}$ (s)	16.5 ^a^	16.4 ^a^	14.8 ^b^
	$\eta_{0}$ (Pa·s× 10^6^)	2.1 ^b^	2.9 ^b^	3.5 ^a^
	$J_{r0}$ (Pa^−1^ × 10^−5^)	7.1 ^a^	2.4 ^b^	2.9 ^b^
	$J_{r1}$ (Pa^−1^ × 10^−5^)	4.1 ^a^	1.6 ^b^	1.8 ^b^
	$t_{r1}$(s)	2.1 ^a^	1.5 ^b^	1.7 ^b^
	$J_{r2}$ (Pa^−1^ × 10^−5^)	4.5 ^a^	1.8 ^b^	2.2 ^b^
	$t_{r2}$ (s)	34.0 ^a^	31.8 ^b^	31.1 ^b^
**Frequency sweep test of batter**	G′ (Pa × 10^3^)	2.2 ^c^	4.3 ^b^	4.6 ^a^
	G″ (Pa × 10^3^)	0.6 ^b^	1.2 ^a^	1.2 ^a^
**Texture profile analysis of roll**	Hardness (N)	312.3 ^c^	445.2 ^b^	796.1 ^a^
	Springiness (m)	2.4 ^a^	1.4 ^b^	1.1 ^c^
	Cohesiveness	0.7 ^a^	0.7 ^a^	0.6 ^b^
	Gumminess (N)	220.7 ^c^	305.1 ^b^	506.0 ^a^
	Resilience	0.4 ^a^	0.4 ^a^	0.4 ^a^
**Baking performance**	Volume (cm^3^)	330.0 ^a^	315.0 ^b^	283.3 ^c^
	SV (cm^3^/g)	3.0 ^a^	2.7 ^b^	2.4 ^c^
	YL (%)	14.1 ^a^	11.8 ^b^	10.7 ^c^
**Color of batter**	L*	81.4 ^a^	72.7 ^b^	75.9 ^b^
	a*	3.0 ^b^	5.6 ^a^	5.5 ^a^
	b*	23.2 ^b^	33.9 ^a^	31.7 ^a^
**Color of roll**	L*	77.8 ^a^	72.2 ^b^	69.4 ^b^
	a*	1.8 ^b^	5.7 ^a^	6.6 ^a^
	b*	29.8 ^b^	36.6 ^a^	37.6 ^a^
**Sensory evaluation**	Appearance	6.4 ^a^	6.5 ^a^	6.4 ^a^
	Taste	7.1 ^a^	7.2 ^a^	7.2 ^a^
	Texture	7.5 ^a^	7.3 ^b^	7.2 ^b^
	Overall quality	6.9 ^a^	7.0 ^a^	7.0 ^a^

* Means (n = 3) followed by different letters within a row are significantly different at *p* < 0.05 (Tukey test). Creep-recovery test; *J_max_*: maximum compliance during creep, *J_final_*: final compliance during recovery, RCY: recoverability, J-Jr: flowability, *J*_c0_: instantaneous elastic compliance during creep phase, *J*_c1_ and *J*_c2_: retarded elastic compliance during creep phase, *t*_1_ and *t*_2_: retardation time during creep phase, *η*_0_: pure viscosity, *J*_r0_: instantaneous elastic compliance during recovery phase, *J*_r1_ and *J*_r2_: retarded elastic compliance during recovery phase, and *t*_r1_ and *t*_r2_: retardation time during recovery phase. Frequency sweep test; G’: solid-like behavior, and G”: liquid-like behavior. Baking performance; SV: Specific volume, and YL: Yield loss.

## Data Availability

The datasets generated for this study are available on request to the corresponding author.

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
