# Peer review of "Modeling the Influence of Okara Flour Supplementation from Time-Temperature Drying Treatment on the Quality of Gluten-Free Roll Produced from Rice Flour"

_foods, 2023, doi:10.3390/foods12183421_

Round 1

Reviewer 1 Report

The manuscript ID: foods-2599067, entitled "Modelling the influence of okara flour supplementation on the quality of gluten-free roll produced from rice flour" is well written and presents well designed experiment. The research is quite interesting due to the fact that the gluten-free food sector can be one of the most profitable branches of the food industry, not only because of people suffering from celiac disease, but also because of the prevailing trend where a significant number of people switch to a gluten-free diet, which is an expression of the search for alternative ways of eating compared to the traditional one, which also justifies the purposefulness of the research undertaken.

The abstract is written correctly.

The Introduction section include an adequately argumentation of the research goal. 

Well-chosen literature is cited, but more recent references need to be added.

The material and methods are described sufficiently.

Interesting results were obtained by suitable methods, but the discussions must be improved with the latest references supporting the results of the study.

The conclusions were supported by the results, and respond to the main objectives of the work.

Provided literature is relevant to the research, but the references from recent five years need to be added.

Manuscript can be further improved taking following points into consideration.

Line 60: in vitro instead of in vitro

Line 120: with instead of with

Line 138: The sample was prepared with yeast addition?

Line 147: Compliance instead of Compliance

Line 161, 162: Please verify the equations, “exp” was write two time.

Line 227: … total dietary fiber (1.32 times, dry basis) … Please verify!

Line 262: please verify the Eq. 4.  Correct is: RCY (%) = (?????? / ?max)×100

Please insert PCA biplot as Figure 3 instead of the graph for storage (G’) and loss (G”) modulus as a function of frequency (ω).

Author Response

The abstract is written correctly.

Reply: Thank you reviewer for the comment.

The Introduction section include an adequately argumentation of the research goal. 

Reply: Thank you reviewer for the comment.

Well-chosen literature is cited, but more recent references need to be added.

Reply: Modifications were made – they are in red font.

The material and methods are described sufficiently.

Reply: Thank you reviewer for the comment.

Interesting results were obtained by suitable methods, but the discussions must be improved with the latest references supporting the results of the study.

Reply: Modifications were made – they are in red font.

The conclusions were supported by the results, and respond to the main objectives of the work.

Reply: Thank you reviewer for the comment.

Provided literature is relevant to the research, but the references from recent five years need to be added.

Reply: Modifications were made – they are in red font.

Manuscript can be further improved taking following points into consideration.

Line 60: in vitro instead of in vitro

Reply: Modifications were made – they are in red font.

Line 120: with instead of with

Reply: Modifications were made – they are in red font.

Line 138: The sample was prepared with yeast addition?

Reply: Modifications were made – they are in red fontRheological measurement of gluten-free roll batter without yeast addition was conducted by frequency sweep test using a rheometer AR-1000N (TA Instruments, New Castle, DE, USA) equipped with a Peltier plate controlling the temperature to 25 oC.”

Line 147: Compliance instead of Compliance

Reply: Modifications were made – they are in red font.

Line 161, 162: Please verify the equations, “exp” was written two time.

Reply: Modifications were made – they are in red font.

Line 227: … total dietary fiber (1.32 times, dry basis) … Please verify!

Reply: It was verified, and the value is correct. It was calculated by (4.63-1.99)/1.99 = 1.32

Line 262: please verify the Eq. 4.  Correct is: RCY (%) = (?????? / ?max)×100

Reply: Yes, we verify that it is correct.

Please insert PCA biplot as Figure 3 instead of the graph for storage (G’) and loss (G”) modulus as a function of frequency (ω).

Reply: PCA biplot was inserted as Figure 3.

Reviewer 2 Report

The influence of okara flour supplementation on the quality of gluten-free rolls produced from rice flour was studied. The results would promote the utilization of okara and add value to this by-product. Moreover, it also has essential significance for enriching gluten-free food types and enhancing their nutritional value. While the manuscript was well designed, some issues should be addressed. The main concerns:

1.    The effect of time-temperature drying treatments on okara flour was not included in the title. If the authors want to keep this study, they should rewrite the title to include it. If not, they can provided as a supplement document.

2.    The manuscript could have been more concise, and part of the content needs logistics. For example, in the Introduction, what has been done in this study should be listed together, and what has been found in these studies should follow them. To be specified, a detailed method (L15) should not be listed in the Abstract.

3.    With this vision, Table 1 is part of the Results and should not be shown in the part of the Method. The same problem also exists in other parts of the manuscript.

4.    Except for the nutritional value, there needs to be more background on the utilization of okara in gluten-free food development and its effect on food products. We can not get the novelty of this study with the Introduction. Moreover, the authors referred to the IDF several times in the Introduction, whereas they have not been determined in this study nor discussed in the Results and Discussion. So, too much content on the IDF in the Introduction was not appropriate. It is better to rephrase the Introduction to show this study's novelty and significance.

5.    Though data was analyzed with significance analysis, the results were not discussed with significance.

6.     In L119, there is a subtitle 2.3.1 after 2.3. Using a subtitle is inappropriate when there is only one subtitle. Moreover, it’s obvious that method 2.3 and 2.3.1 are two independent methods. Thus, the number of the subtitle should be carefully considered.

7.    In Table 3, the color of the batter was shown, whereas the authors have not explained how it was determined in the Methods.

8.    Figure 1 is not necessary.

9.    L204, the description of the results could have been more rigorous.

10. How PCA was conducted in this study should be described. Data discussed in 3.4 can not be found in Figure 3. Neither data in Figure 3 can not be found in 3.4. Figure 3 is the result of the frequency sweep test. It is weird to be shown as part of the result of the PCA. The authors should pay particular attention here.

11. Okara may have super water absorption capacity due to its high DF content. Its high water absorption capacity would play a role in the batter's rheological properties and the gluten-free roll's properties. Thus, the discussion should take it into account.

 Moderate editing of English language is required.

Author Response

The influence of okara flour supplementation on the quality of gluten-free rolls produced from rice flour was studied. The results would promote the utilization of okara and add value to this by-product. Moreover, it also has essential significance for enriching gluten-free food types and enhancing their nutritional value. While the manuscript was well designed, some issues should be addressed. The main concerns:

  1. The effect of time-temperature drying treatments on okara flour was not included in the title. If the authors want to keep this study, they should rewrite the title to include it. If not, they can provided as a supplement document.

Reply: Modifications were made – they are in red font. “Modeling the influence of okara flour supplementation from time-temperature drying treatment on the quality of gluten-free roll produced from rice flour”

  1. The manuscript could have been more concise, and part of the content needs logistics. For example, in the Introduction, what has been done in this study should be listed together, and what has been found in these studies should follow them. To be specified, a detailed method (L15) should not be listed in the Abstract.

Reply: Modifications were made and they are in red font in the introduction section and “The okara was milled into flour to pass through 60 US mesh screen, and analyzed for chemical composition, antioxidant level and total microbial count.” – was removed.

  1. With this vision, Table 1 is part of the Results and should not be shown in the part of the Method. The same problem also exists in other parts of the manuscript.

Reply: Table 1 has been moved to the Results section.

  1. Except for the nutritional value, there needs to be more background on the utilization of okara in gluten-free food development and its effect on food products. We cannot get the novelty of this study with the Introduction. Moreover, the authors referred to the IDF several times in the Introduction, whereas they have not been determined in this study nor discussed in the Results and Discussion. So, too much content on the IDF in the Introduction was not appropriate. It is better to rephrase the Introduction to show this study's novelty and significance.

Reply: Modifications were made – IDF was removed in introduction and the utilization of okara in gluten-free food development and its effect on food products was added. They are in red. “However, limited research has observed variations in okara’s antioxidant levels during processing. Currently, the demand for gluten-free products, catering to individuals with celiac disease or wheat allergies, is on the rise. In gluten-free products, wheat flour is normally substituted with high-carbohydrate white rice flour. Therefore, in this study, we focus on the processing soybean pulp (okara) to produce a novel functional ingredient in gluten-free rolls.”

  1. Though data was analyzed with significance analysis, the results were not discussed with significance.

Reply: The results were described with significance throughout the manuscript.

  1. In L119, there is a subtitle 2.3.1 after 2.3. Using a subtitle is inappropriate when there is only one subtitle. Moreover, it’s obvious that method 2.3 and 2.3.1 are two independent methods. Thus, the number of the subtitle should be carefully considered.

Reply: Modifications were made – they are in red font.

  1. In Table 3, the color of the batter was shown, whereas the authors have not explained how it was determined in the Methods.

Reply: The method was in section 2.6.1 – “Crumb color of gluten-free roll and batter were also analyzed with a HunterLab Color-Flex (Hunter Associates Inc., Reston, VA, USA). Parameters of L* (brightness; 0: black, 100: white), a* (+a: redness; −a: greenness) and b* (+b: yellowness; −b: blueness) values were recorded [17]. Measurements were recorded on three replicates.”

  1. Figure 1 is not necessary.

Reply: Figure 1 was removed from the manuscript.

  1. L204, the description of the results could have been more rigorous.

Reply: The results were described with significance throughout the manuscript.

  1. How PCA was conducted in this study should be described. Data discussed in 3.4 cannot be found in Figure 3. Neither data in Figure 3 cannot be found in 3.4. Figure 3 is the result of the frequency sweep test. It is weird to be shown as part of the result of the PCA. The authors should pay particular attention here.

Reply: Modifications were made – they are in red font.

  1. Okara may have super water absorption capacity due to its high DF content. Its high water absorption capacity would play a role in the batter's rheological properties and the gluten-free roll's properties. Thus, the discussion should take it into account.

Reply: Modifications were made – they are in red font. "The high DF content in Okara suggests a high-water absorption capacity and thus influencing the batter's rheological properties and the gluten-free roll's properties"  

Round 2

Reviewer 2 Report

The authors have revised the manuscript with regard to all the comments and suggestions provided before.